# CHEcking Diagnostic Differential Ability of Real Baseline Variables and Frailty Scores in Tolerance of Anti-Cancer Systemic Therapy in OldEr Patients (CHEDDAR-TOASTIE)

**DOI:** 10.3390/cancers17203303

**Published:** 2025-10-13

**Authors:** Helen H. L. Ng, Isa Mahmood, Francis Aggrey, Helen Dearden, Mark Baxter, Kieran Zucker

**Affiliations:** 1Leeds Teaching Hospitals NHS Trust, Leeds LS9 7TF, UK; 2 Faculty of Medicine and Health, School of Medicine, University of Leeds, Leeds LS2 9JT, UK; 3Division of Molecular and Clinical Medicine, University of Dundee Division of Medical Sciences, Dundee DD1 4HN, UK; 4Tayside Cancer Centre, Tayside University Hospitals NHS Trust, Dundee DD3 8EA, UK

**Keywords:** older population, chemotherapy-related toxicities, prediction models, chemotherapy

## Abstract

Older adults are more prone to severe side effects (toxicities) from chemotherapy. The initial observational study found that scoring systems used to predict toxicities in a 65+ UK population receiving first-line chemotherapy performed poorly. This subsequent study aims to explore whether additional frailty and baseline health data can improve the performance of toxicity prediction models. Data from the observational study were used: factors such as age, sex, weight, a patient’s own assessment of health (AHQ), and number of comorbidities were analyzed for their predictive performance. Then, predictive models were built using various statistical and machine learning methods. Among 322 patients, 22% had toxicities. Ten factors were weakly linked to toxicities, including AHQ and a high baseline neutrophil count. The best performance predictive models had only low–moderate accuracies, insufficient for clinical use in predicting toxicities. Further research is needed to develop a more robust predictive scoring system.

## 1. Introduction

The United Kingdom (UK) has an aging population, with the number of adults aged 65 years and older estimated to increase by a third to 10 million by 2040 [1]. This aging population, combined with better diagnostic pathways and treatment outcomes, is driving a growing percentage of the population within this age bracket to live with and beyond cancer [2]. Chemotherapy remains a core element of many curative and palliative treatments, but previous studies have demonstrated that older patients are more likely to experience treatment-related toxicities [3,4]. Despite this, robust mechanisms for the identification of older patients who will develop significant toxicity remain elusive [5]. This increased risk of significant toxicity has resulted in altered clinical practice in many settings, with older patients often receiving less intensive treatment either by reducing doses or limiting the number or types of systemic treatments offered [6,7]. This results in a situation where some patients who would have tolerated more aggressive treatment are undertreated, potentially limiting the benefits of their chemotherapeutic interventions. Other patients, however, will still develop significant toxicity, negatively impacting their quality of life to a greater extent, limiting or nullifying the benefits they gain from this treatment [8,9].

The evidence base for the management of older patients is lacking as many randomized controlled trials exclude older patients due to age, frailty or presence of comorbidities [2,6]. Results from younger, fitter patients are therefore extrapolated onto an older and often multimorbid population [10]. Providing tailored information to support clinician and patient decision making in older patients is therefore often a challenge, as outcomes in terms of therapeutic response or side effects can vary significantly [11]. Practice in how to manage older patients with cancer often varies across geographies and is often pointed to as a potential explanation for significant outcome differences in cancer when comparing the UK to other high-income countries [12]. A review from the International Society of Geriatric Oncology has found that there is a lack of representation of older adults across European guidelines [13]. This is a recognized issue in the UK, which the recent Royal College of Radiologists (RCR) guidelines aim to address by making assessment and management for patients’ frailty an essential part of cancer care [14].

Over the past decade, research around the concepts of multi-morbidity and frailty has had growing interest [15,16]. Within the oncology setting, several predictive tools have been developed to support the assessment of the risk of chemotherapy toxicity, including the Cancer and Aging Research Group (CARG) score [5] and the Chemotherapy Risk Assessment Scale for High-Age Patients (CRASH) Score [17]. Despite their use in some clinical settings, their evidence base is variable [12], and the level of accuracy has been shown within the TOASTIE study to be limited in a UK population of older patients receiving first-line chemotherapy [18].

Given the lack of consistent evidence, this raises the question of whether it is feasible to use baseline demographic data within a bespoke scoring system to predict toxicity to a clinically useful level. This study aims to answer this question by evaluating the baseline characteristics and frailty assessment information for predicting whether older individuals will develop high-grade toxicity when treated with chemotherapy.

## 2. Materials and Methods

### 2.1. Data Source

This study makes use of data previously collected within the TOASTIE study [18,19,20], a prospective observational study which included 339 patients recruited across 18 NHS institutions between 2021 and 2022. Included patients were over the age of 65 prior to commencing first-line chemotherapy for a solid-organ malignancy with any intent. Full information about the study protocol can be found within their published protocol paper [20].

Anonymized data from the TOASTIE study were used for analysis and predictive model building in this study.

### 2.2. Data Collection and Items

The TOASTIE dataset [20] included data items collected at baseline. These included demographics, tumor diagnostic information, treatment information, a questionnaire about the patient’s own health assessment, Rockwood Clinical Frailty Score (CFS) [21] as part of the Additional Health Questionnaire (AHQ) and researcher-estimated risks of toxicities (using the CARG score [5]). The outcome of interest was the occurrence of severe (grade 3+) chemotherapy-related toxicity in each case, as defined by the National Cancer Institute Common Terminology Criteria for Adverse Events v5 [22]. A list of all data items within the dataset can be found in Appendix A.

### 2.3. Data Analysis

All data analysis was undertaken using R version 4.3.1 and standard packages available on CRAN. A list of the non-base R packages can be found in Appendix A.

#### 2.3.1. Inclusion/Exclusion Criteria

All patients meeting the initial TOASTIE study inclusion criteria were included. Patients for whom the outcome (grade 3–5 chemotherapy-related toxicities) was unavailable were excluded.

#### 2.3.2. Internal Validation of Models

Prior to building the predictive models, patients were separated into train, validation and test cohorts in a 70:15:15 ratio. The cohorts were partitioned based on the ratio of presence and absence of the dependent (grade 3+ toxicities). An up-sampled training set was generated to address the class imbalance in the original dataset. Instances of patients with grade 3+ toxicities were oversampled until there was a balanced number of instances of “toxicity” and “no toxicity”. This was only performed for the training data, while the validation and test data were left unchanged.

#### 2.3.3. Handling Missing Data

The patterns and scale of missingness were assessed for each variable within the dataset. For variables assumed missing at random, Multivariate Imputation by Chained Equations (MICE) imputation was implemented. MICE imputation was applied on both the original and up-sampled training sets. These were referred to as “imputed training sets”. Complete case analysis was also applied to the original and up-sampled training sets, these will be referred to as the “complete case training sets”.

#### 2.3.4. Selecting Variables

##### Candidate Data Items

Data items included in the analysis were taken from the TOASTIE trial dataset. The original dataset included 40 demographics, self-assessment, and researcher’s assessment variables. A full list can be found in Appendix A.

##### Variable Selection

Two approaches were applied to identify relevant variables for inclusion within the subsequent modeling strategies: by clinical knowledge and by using statistical associations.

Four medical doctors including one cancer specialist reviewed the data items available within the dataset. They developed a consensus of variables to be used based on their clinical knowledge and expertise.

A second numerical approach to variable selection was conducted where Spearman’s correlation was used to identify associations between continuous variables. Chi-square (χ^2^) or Fisher’s exact test was used for categorical variables depending on the number of samples. When a particular observation type is small (<5), Fisher’s exact test was applied. Variables were tested against presence of grade 3+ chemotherapy-related toxicities for associations. Statistical significance was considered at *p* < 0.05.

#### 2.3.5. Building Predictive Models

Three methods were used, including logistic regression, LASSO and random forest. All three methods were applied on the eight training sets generated above (Figure 1).

##### Logistic Regression

Logistic regression was carried out with glm from the stats package with the binomial family. Two formulas were built: one based on the clinically informed variables and the other based on the statistically significant variables. The training dataset was used to build predictive models with each formula. The validation set was used to find the optimal cut point for classification. The method of cut point estimation was by maximizing the sum of sensitivity and specificity. The testing set applied the said cut point and produced the results. The model performance metrics will be reported from all sets for evaluation.

##### LASSO

Variables were treated as numeric for LASSO-logistic regression. LASSO-logistic regression models were trained by the training datasets for the clinically informed variables and separately for the statistically significant variables, as illustrated in Figure 1. To determine the optimal regularization parameter (“best lambda”), the validation dataset was used. This served to balance the trade-off between bias and variance, minimizing the overall error to optimize the model’s performance. Subsequently, the models were tested with the best lambda value on the test dataset to assess the predictive accuracy.

##### Random Forest

In order to develop the random forest models, the “ranger” package from R was used. To optimize the predictive performance of the random forest models that were developed, a systematic approach to hyperparameter tuning was employed. This involved first creating a tuning grid, which consisted of key hyperparameters, including the following:
Mtry: This parameter determines the number of variables randomly sampled at each split in the decision trees of the random forest. Values of 2, 4, and 6 were tested to evaluate different feature subset sizes.splitrule: Two splitting rules, “gini” and “extratrees”, were evaluated to determine the optimal method for partitioning nodes in the decision trees.min.node.size: This parameter specifies the minimum number of observations required to create a terminal node in the tree. Values of 1, 3, and 5 were selected to assess model sensitivity to node size.

Following the establishment of the tuning grid, each model configuration was trained and evaluated using five-fold cross-validation. This was performed to reduce the risk of overfitting by helping to provide a reliable estimate of the model’s performance on unseen data.

Further tuning was then conducted using several combinations of tree numbers. Each model was built with 50 trees for consistency and to prevent overfitting due to the relatively small datasets.

After completing the tuning process, the optimal set of hyperparameters was identified based on maximizing accuracy across the cross-validated folds. Using the best-tuned hyperparameters for each data split, the final random forest models were constructed using the entire training dataset. These models were then tested on the “test” datasets.

#### 2.3.6. Model Performance Evaluation

For logistical regression and LASSO, the performances of the models were optimized using the validation data, by finding the optimal cut point and best lambda values, respectively; holdout testing data was used to assess the model’s robustness and generalizability.

In the case of random forest, cross-validation was employed during the training process to tune the hyperparameters, and holdout testing data was then used to assess the model’s generalizability.

Performance metrics of each of the models, including accuracy with a 95% confidence interval, *p*-value, sensitivity (or recall), specificity, positive predictive value (PPV) (or precision), negative predictive value (NPV) and balanced accuracy were reported. Receiver Operating Characteristic (ROC) curves were plotted to visualize the trade-offs between sensitivity and specificity for different threshold values. The area under the curve (AUC) was then determined for each ROC curve to give a measure of overall performance.

## 3. Results

### 3.1. Demographic Results

A total of 322 (92.8% of TOASTIE data) patients were included. In total, 164 (50.9%) were males. The incidence of Grade 3+ toxicities was 22.0% (71/322). The baseline characteristics of the patients can be found in Table 1. Most (274, 85%) patients were of good Performance Status (WHO Performance Status 0 or 1), and similar numbers (276, 86%) lived with very mild frailty (Rockwood CFS ≤ 4).

Four variables were excluded at this stage, including patient number, as this was irrelevant to the analysis. Height and weight were removed as BMI (Body Mass Index) was included. There was a duplication and/or error in data collection for the number of comorbidities, as the number was not all 0 when it was stated as “no” in the “presence of comorbidities”. Only one column was kept with regard to comorbidities (yes/no). “Presence of psychological issues” was removed due to data quality issues.

Results of the partition of data into training, validation and test cohorts can be found in Appendix A. Each partition has 21–24% of patients with G3+ toxicities.

Missing data is analyzed with results shown in Figure 2. There are five variables where >15% of observations are missing (high creatinine clearance levels, low creatinine clearance levels, cancer stage, presence of metastases and Karnofsky performance score). 27.4% of data was missing with regard to the presence of metastasis. There is no identified pattern to missingness (Figure 2).

The selected clinically informed variables were age, sex, BMI, ECOG Performance Status, cancer type, number of chemotherapy drugs, presence of co-morbidities and number of regular medications.

Table 1 shows the ten variables from the baseline demographics and patient-reported factors that were statistically different between groups with and without toxicities. The variables include the WHO/ECOG Performance Status score; a high count of baseline neutrophils; six measures from the Assessment of Health Questionnaire (interference of social activities, ability to take own medication, effect of health in walking one block, weight loss in the past 3 months, general mobility and decline in food intake); Rockwood Clinical Frailty Score; and patient’s own measure of comparison of own health to others. The full table of demographics and patient factors can be found in Appendix A.

### 3.2. Statistical Associations of Variables

The results from univariate regression and multivariate regression analyses using the ten clinically informed variables can be found in Table 2. Univariable regression analysis independently analyzed each variable for any potential relationship with the dependent (grade 3+ chemotherapy-relatedtoxicities). From the univariable regression, only Rockwood CFS had a statistically significant (*p* < 0.05) coefficient. Multivariable regression analysis was performed with the ten clinically informed variables, of which Performance Status and the number of chemotherapy drugs were statistically significant. Furthermore, after multiple imputation was performed, another multivariable regression analysis with all variables was performed to observe for any complex relationships between all the available variables and the dependent. With a small effect (coefficient of 0.92), the researcher’s estimated risk of significant toxicity in percentage is the only statistically significant variable from the multivariable regression analysis.

### 3.3. Performance of Predictive Models

The model performance metrics on the test dataset can be found in Table 3. Across the different modeling methods, the models with clinically selected variables have a balanced accuracy between 0.4298 and 0.6075, with an AUC between 0.4298 and 0.636. As for the models using identified significant variables, the balanced accuracies range between 0.5724 and 0.6469, with an AUC between 0.5724 and 0.659. All performance metrics, including sensitivity, specificity, negative predictive value and positive predictive value, can be found in Appendix A. The full code for all analyses can be found via the link in Appendix A.

## 4. Discussion

Several models have been developed to predict chemotherapy toxicity, including the CRASH and CARG scores. Despite their common use in clinical settings, their evidence base varies. A study has shown that their predictive performances of overall toxicities were similar, with a ROC-AUC between 0.650 and 0.681 [11]. An Australian study has found that 58% of those classified by the CARG score as low risk, in fact, had severe toxicities [24]. The TOASTIE study also found limited accuracy for application in a UK population of older patients [17]. This prompts the question of whether using baseline frailty data in a precise scoring system is feasible for predicting toxicity. The tolerance of chemotherapy in older cancer patients is a concern, as predicting the risk of chemotherapy toxicity in advance can help clinicians identify vulnerable populations, allowing for more personalized treatment plans.

This current study has reused the data collected as part of a prospective UK multicenter study, with attempts to develop an objective predictive model based on an older cohort. Despite various model-building methods, results were at best only marginally better than chance, with the balanced accuracy of most models hovering around 60%. For example, a model built with logistic regression using complete cases, with variables determined by statistical associations, achieved a balanced accuracy of 64% and a negative predictive value (NPV) of 87%. While models with low balanced accuracy may still have clinical utility, such as reliably ruling out toxicity, this logistic regression model is particularly promising with its high accuracy in predicting no-toxicity. The model’s ability to accurately predict no toxicity allows clinicians to identify patients unlikely to experience adverse effects from standard dosing. This may support more confident decision making to avoid unnecessary dose reductions, which can compromise therapeutic efficacy, and allow patients to begin treatment at full dose. Furthermore, a high NPV model provides a strong basis for clinician–patient discussions regarding treatment safety. When used in combination with the Comprehensive Geriatric Assessment by the International Society of Geriatric Oncology (SIOG), patients can be reassured when the model predicts low risk and scores are low, potentially improving adherence and reducing anxiety [2]. It is interesting to note that the top predictors of the statistically identified variables used in this model include WHO Performance Status and patient-reported interference of social activities due to health—the clinical significance of this is unclear and requires further investigation. Furthermore, it would be prudent to collect larger cohorts to enable analysis of any differences between patients who had initial dose reductions and patients who subsequently had dose reductions.

Previously published research suggested that there is clinical utility with the use of baseline data in predicting high-grade toxicity in older patients receiving chemotherapy [25] or for the use of risk prediction for specific cancers [26]. The results of this study, however, provide insufficient evidence for this. Across the multiple combinations of variables and modeling strategies applied, no models were able to robustly predict adverse toxicity outcomes (Table 3). The performance of the models showed an imbalance between PPV and NPV. Even the statistical association between individual variables and outcomes was extremely limited. These results suggest that the baseline variables collected within the TOASTIE study do not explain enough of the variation around the mean to enable robust predictions. However, it is worth noting the results from the univariate analysis (Table 1) as the statistically significant variables do correlate with the previous literature [27,28,29,30]. Further research should also explore clinicians’ intuition, as measured by the “Researcher’s estimated risk of significant toxicity in percentage”, which may play a role in a prediction model (Table 2). It would also be interesting to investigate how clinicians’ length of experience affects their estimated risks of toxicities for a patient.

When considering the potential value of a clinical prediction tool, it is important to assess the threshold at which the performance enables the output to become clinically actionable. While patient data from different hospitals could potentially provide external validation, considering local patient sets or specific cancer types might yield better results. Additionally, the variability in chemotherapy regimens—such as dosage and drug combinations—may not be fully accounted for in the current models. These would be potential areas for further development in future research; however, they would require much larger datasets to represent these sub-cohorts at a sufficient scale for any patterns to emerge.

In our study, several limitations need consideration. First, the sample size was modest in size, and missing data posed challenges despite using imputation techniques like MICE. The incidence of grade 3–5 toxicities within our dataset was relatively low (just over 20%) in the whole cohort, which led to class imbalance issues that could impact model performance, especially in terms of sensitivity and specificity for the minority class. Although this was dealt with using methods such as up-sampling, this might have introduced bias or overfitting. Up-sampling is a technique that involves duplicating minority class samples, but it can mislead the model’s learning, resulting in poorer performance on unseen data. While up-sampling was employed to address class imbalance, it inherently carries the risk of overfitting, particularly when the model begins to memorize duplicated instances of the minority class rather than learning generalizable patterns. This can lead to inflated performance during training but poorer generalization to unseen data. To mitigate this, up-sampling was strictly confined to the training dataset, ensuring that the test data remained untouched and unbiased. Furthermore, model validation using separate test sets helped assess true predictive performance and reduce the likelihood of overfitting-induced optimism in reported metrics. Evaluation bias has been minimized in our methods as up-sampling is performed only on the training dataset, not the test dataset. This avoids inflated performance metrics due to data leakage. Further, our models were constructed using a specific set of variables, potentially excluding other important predictors of chemotherapy toxicity. Although two methods were applied, including using clinical knowledge and then with statistically significant data, there might be other combinations of variables, which could improve the model. A brute force approach involves systematically exploring all possible combinations of variables without any optimization or shortcuts, potentially identifying interactions and predictors that were otherwise overlooked. This method was considered the last option to theoretically be able to exhaust all other possible outcomes for a possible model. However, early in the study, it was apparent that there was insufficient computational power to complete this analysis and thus remains a limitation to the study. Despite this, our results (Table 3) remain supportive in that baseline frailty data are insufficient to provide clinical use in predicting risk of toxicity. Conversely, it is important to note that there is potential to design a no-toxicity prediction for identifying low-risk patients. As shown by the slightly higher NPV from one of the modeling strategies, it seems that data of this type is more suited to predictions of said class rather than the minority class. It is prudent to use larger datasets in the future to aim for a model with a clinically useful NPV.

## 5. Conclusions

Variables collected from CARG or at the patient’s baseline lack robust clinical utility to guide treatment direction based on the risk of chemotherapy-related toxicities. Several methods have been trialed, and none have yielded robust results. Clinicians should carefully reconsider the use of CARG or other variables when assessing a patient’s risk of developing severe chemotherapy-related toxicity. However, there is scope to enhance the prediction of no toxicity, which in turn can identify lower-risk patients who may not require dose reductions, potentially improving overall outcomes. Further research into other baseline variables is required for building a more robust predictive scoring system.

## Figures and Tables

**Figure 1 cancers-17-03303-f001:**
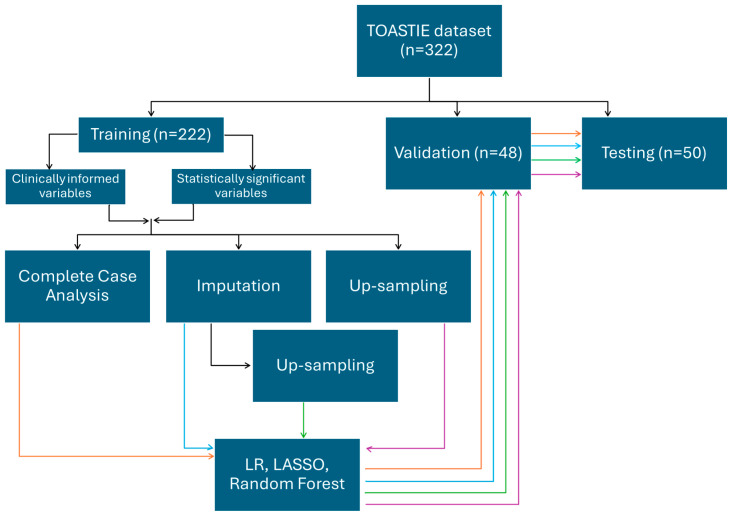
An infographic demonstrating the splitting of data into a train/validation/testing set. Using the training set, two combinations of variables and four different methods of processing are chosen. The different training sets were each used with logistic regression, LASSO and random forest for model building, which are then internally validated. Results are then reported from the testing dataset. Orange follows the route of training datatset which have been analyssed as complete case; blue by imputation; green by upsampling after imputation and purple by upsampling only.

**Figure 2 cancers-17-03303-f002:**
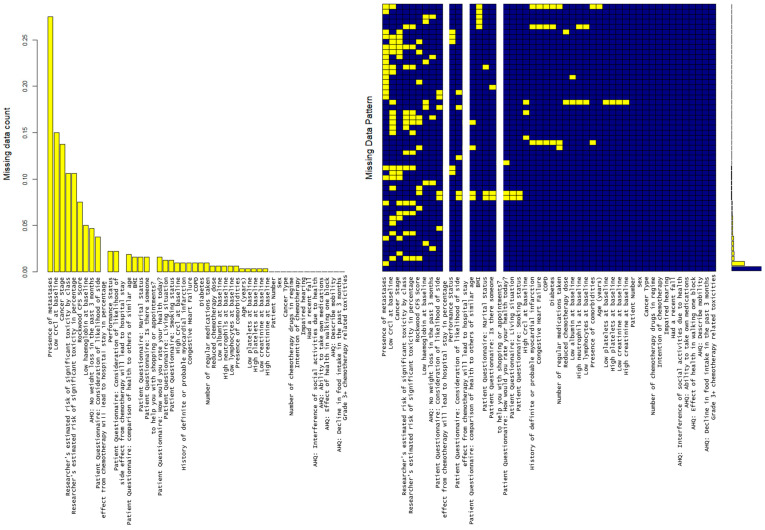
MICE plot showing that there are 5 variables where 15+% of observations are missing (“crcl_high”, “crcl_low”, “ca_stage”, “mets”, “karnofsky_ps”), with the highest being presence/absence of metastasis (27.4%). The md.pattern plot shows the number of missing data. Yellow blocks show the missing data visually in comparison to available data in blue.

**Table 1 cancers-17-03303-t001:** Baseline characteristics of the study participants, highlighting the variables that are significantly different, as measured by the chi-square (**x**^2^) between groups with and without severe chemotherapy-related toxicities. The full table of baseline characteristics can be found in Appendix A. Abbreviations: WHO/ECOG [23]—World Health Organization/Eastern Cooperative Oncology Group), AHQ—Assessment of Health Questionnaire; Rockwood CFS—Rockwood Clinical Frailty Score [21].

*Dependent: Severe Chemotherapy-Related Toxicities*		Grade ≤ 2 Toxicity	GradeRADE 3+ Toxicity	Total	x^2^ (*p*)
Total N (%)		251 (78.0)	71 (22.0)	322	
WHO/ECOG Performance Status (0–4)	Mean (SD)	0.7 (0.7)	1.0 (0.7)	0.8 (0.7)	0.004
High baseline neutrophils	FALSE	208 (83.9)	50 (71.4)	258 (81.1)	0.03
	TRUE	40 (16.1)	20 (28.6)	60 (18.9)	
AHQ: interference of social activities due to health	All the time	10 (4.0)	10 (14.1)	20 (6.2)	0.004
	Most of the time	28 (11.2)	9 (12.7)	37 (11.5)	
	Some of the time	38 (15.1)	15 (21.1)	53 (16.5)	
	A little of the time	21 (8.4)	8 (11.3)	29 (9.0)	
	None of the time	153 (61.0)	28 (39.4)	181 (56.2)	
	(Missing)	1 (0.4)	1 (1.4)	2 (0.6)	
AHQ: ability to take own medications	Without help	240 (95.6)	62 (87.3)	302 (93.8)	0.035
	With some help or reminders	9 (3.6)	6 (8.5)	15 (4.7)	
	Unable	1 (0.4)	2 (2.8)	3 (0.9)	
	(Missing)	1 (0.4)	1 (1.4)	2 (0.6)	
AHQ: effect of health on walking one block	Limited a lot	10 (4.0)	6 (8.5)	16 (5.0)	0.001
	Limited a little	21 (8.4)	16 (22.5)	37 (11.5)	
	No limitations	219 (87.3)	48 (67.6)	267 (82.9)	
	(Missing)	1 (0.4)	1 (1.4)	2 (0.6)	
AHQ: weight loss in the past 3 months	Yes	130 (51.8)	48 (67.6)	178 (55.3)	0.004
	No	111 (44.2)	16 (22.5)	127 (39.4)	
	(Missing)	10 (4.0)	7 (9.9)	17 (5.3)	
AHQ: describe mobility	Bed- or chair-bound	1 (0.4)	0 (0.0)	1 (0.3)	<0.001
	Can get out, but does not	7 (2.8)	11 (15.5)	18 (5.6)	
	Gets out	242 (96.4)	59 (83.1)	301 (93.5)	
	(Missing)	1 (0.4)	1 (1.4)	2 (0.6)	
AHQ: decline in food intake in the past 3 months	Severe decrease	36 (14.3)	15 (21.1)	51 (15.8)	0.004
	Moderate decrease	76 (30.3)	32 (45.1)	108 (33.5)	
	No decrease	138 (55.0)	23 (32.4)	161 (50.0)	
	(Missing)	1 (0.4)	1 (1.4)	2 (0.6)	
Rockwood CFS	Mean (SD)	2.5 (1.2)	3.0 (1.3)	2.6 (1.2)	0.004
Patient questionnaire: comparison of health to others of similar age	Not as good	24 (9.6)	14 (19.7)	38 (11.8)	0.007
	As good	102 (40.6)	19 (26.8)	121 (37.6)	
	Better	111 (44.2)	31 (43.7)	142 (44.1)	
	Does not know	7 (2.8)	6 (8.5)	13 (4.0)	
	(Missing)	7 (2.8)	1 (1.4)	8 (2.5)	

**Table 2 cancers-17-03303-t002:** Regression analysis of all variables: (1) univariable, (2) multivariable using the clinically selected variables and (3) multivariable with multiple imputation for all available variables. Please note the broad confidence intervals or infinity/NA values in some of the odds ratios (e.g., of high bilirubin or of some types of cancer), which are likely due to instability of the small subgroups. Abbreviations: OR—Odds Ratio; BMI—Body Mass Index; WHO/ECOG [23]—World Health Organization/Eastern Cooperative Oncology Group; AHQ—Assessment of Health Questionnaire; Rockwood CFS—Rockwood Clinical Frailty Score [21]; COPD—Chronic Obstructive Pulmonary Disease; CARG—Cancer and Aging Research Group.

*Dependent: Grade 3+ Toxicities*		*FALSE*	*TRUE*	*OR (Univariable)*	*OR (Multivariable)*	*OR (Multiple Imputation)*
*Age*	Mean (SD)	72.6 (4.9)	72.7 (4.8)	1.00 (0.94–1.07, *p* = 0.899)	1.00 (0.92–1.08, *p* = 0.996)	0.80 (0.63–1.03, *p* = 0.078)
*Sex*	Female	92 (52.3)	23 (46.9)	-	-	-
	Male	84 (47.7)	26 (53.1)	1.24 (0.66–2.35, *p* = 0.509)	1.28 (0.57–2.96, *p* = 0.551)	0.89 (0.14–5.72, *p* = 0.901)
*BMI*	Mean (SD)	27.1 (5.4)	26.8 (5.5)	0.99 (0.93–1.05, *p* = 0.703)	0.98 (0.92–1.05, *p* = 0.595)	1.06 (0.88–1.27, *p* = 0.540)
*ECOG Performance Status (0–4)*	Mean (SD)	0.8 (0.7)	0.9 (0.6)	1.49 (0.93–2.39, *p* = 0.095)	1.94 (1.09–3.55, *p* = 0.027)	2.27 (0.40–12.97, *p* = 0.346)
*Cancer Type*	Upper GI	45 (25.6)	12 (24.5)	-	-	-
	Gynecological	28 (15.9)	5 (10.2)	0.67 (0.20–2.02, *p* = 0.492)	0.69 (0.15–2.96, *p* = 0.628)	0.13 (0.00–3.75, *p* = 0.227)
	Lung	12 (6.8)	2 (4.1)	0.62 (0.09–2.72, *p* = 0.571)	0.57 (0.08–2.74, *p* = 0.521)	0.04 (0.00–2.86, *p* = 0.139)
	Breast	15 (8.5)	6 (12.2)	1.50 (0.46–4.61, *p* = 0.486)	2.28 (0.57–8.91, *p* = 0.233)	3.89 (0.24–62.61, *p* = 0.335)
	Lower GI	36 (20.5)	10 (20.4)	1.04 (0.40–2.69, *p* = 0.933)	1.62 (0.54–4.94, *p* = 0.391)	0.67 (0.07–6.70, *p* = 0.730)
	HPB	9 (5.1)	6 (12.2)	2.50 (0.72–8.42, *p* = 0.139)	1.97 (0.52–7.12, *p* = 0.303)	0.99 (0.04–24.60, *p* = 0.996)
	Urological	20 (11.4)	6 (12.2)	1.13 (0.35–3.34, *p* = 0.836)	1.99 (0.53–7.13, *p* = 0.294)	0.26 (0.01–5.52, *p* = 0.385)
	Other	11 (6.2)	2 (4.1)	0.68 (0.10–3.01, *p* = 0.646)	1.00 (0.13–5.01, *p* = 0.996)	0.13 (0.00–12.95, *p* = 0.372)
*Cancer Stage*	1	9 (5.8)	1 (2.2)	-	-	-
	2	25 (16.2)	6 (13.0)	2.16 (0.31–43.74, *p* = 0.502)	-	39.54 (0.14–11,309.76, *p* = 0.195)
	3	52 (33.8)	19 (41.3)	3.29 (0.56–62.70, *p* = 0.274)	-	125.65 (0.37–42,966.83, *p* = 0.101)
	4	68 (44.2)	20 (43.5)	2.65 (0.46–50.30, *p* = 0.369)	-	18.25 (0.02–14,521.60, *p* = 0.374)
*Presence of metastases*	FALSE	44 (35.2)	16 (39.0)	-	-	-
	TRUE	81 (64.8)	25 (61.0)	0.85 (0.41–1.78, *p* = 0.658)	-	1.38 (0.10–18.85, *p* = 0.806)
*Number of chemotherapy drugs*	Mean (SD)	2.1 (0.6)	2.2 (0.6)	1.64 (0.98–2.77, *p* = 0.059)	2.01 (1.04–4.09, *p* = 0.043)	2.50 (0.52–12.05, *p* = 0.250)
*Intention of chemotherapy*	Neo-adjuvant	52 (29.5)	14 (28.6)	-	-	-
	Adjuvant	42 (23.9)	11 (22.4)	0.97 (0.39–2.36, *p* = 0.951)	-	0.35 (0.04–2.93, *p* = 0.328)
	Palliative	82 (46.6)	24 (49.0)	1.09 (0.52–2.33, *p* = 0.826)	-	3.23 (0.19–53.57, *p* = 0.409)
*Dose reduction*	FALSE	115 (65.3)	27 (55.1)	-	-	-
	TRUE	61 (34.7)	22 (44.9)	1.54 (0.80–2.92, *p* = 0.191)	-	1.58 (0.35–7.04, *p* = 0.544)
*Low baseline hemoglobin*	FALSE	112 (66.3)	23 (52.3)	-	-	-
	TRUE	57 (33.7)	21 (47.7)	1.79 (0.91–3.52, *p* = 0.088)	-	4.54 (0.31–66.74, *p* = 0.249)
*Low platelet count at baseline*	FALSE	169 (96.0)	48 (98.0)	-	-	-
	TRUE	7 (4.0)	1 (2.0)	0.50 (0.03–2.92, *p* = 0.525)	-	0.02 (0.00–1.71, *p* = 0.083)
*High platelet count at baseline*	FALSE	147 (83.5)	43 (87.8)	-	-	-
	TRUE	29 (16.5)	6 (12.2)	0.71 (0.25–1.71, *p* = 0.471)	-	0.14 (0.01–1.79, *p* = 0.131)
*Low creatinine at baseline*	FALSE	141 (80.1)	36 (73.5)	-	-	-
	TRUE	35 (19.9)	13 (26.5)	1.45 (0.68–2.99, *p* = 0.317)	-	2.24 (0.35–14.23, *p* = 0.386)
*High creatinine at baseline*	FALSE	159 (90.3)	41 (83.7)	-	-	-
	TRUE	17 (9.7)	8 (16.3)	1.82 (0.70–4.41, *p* = 0.194)	-	4.62 (0.37–57.69, *p* = 0.231)
*Low creatinine clearance at baseline*	FALSE	18 (12.2)	5 (11.1)	-	-	-
	TRUE	129 (87.8)	40 (88.9)	1.12 (0.41–3.55, *p* = 0.838)	-	0.32 (0.01–12.33, *p* = 0.518)
*High bilirubin count at baseline*	FALSE	167 (94.9)	48 (100.0)	-	-	-
	TRUE	9 (5.1)		0.00 (NA–∞, *p* = 0.985)	-	0.00 (0.00–Inf, *p* = 0.993)
*Low albumin count at baseline*	FALSE	130 (74.3)	34 (69.4)	-	-	-
	TRUE	45 (25.7)	15 (30.6)	1.27 (0.62–2.52, *p* = 0.494)	-	0.67 (0.10–4.29, *p* = 0.668)
*High neutrophil count at baseline*	FALSE	147 (83.5)	36 (73.5)	-	-	-
	TRUE	29 (16.5)	13 (26.5)	1.83 (0.85–3.82, *p* = 0.114)	-	8.38 (0.90–77.83, *p* = 0.061)
*Low lymphocyte count at baseline*	FALSE	139 (79.0)	39 (79.6)	-	-	-
	TRUE	37 (21.0)	10 (20.4)	0.96 (0.42–2.05, *p* = 0.925)	-	1.19 (0.17–8.26, *p* = 0.856)
*Presence of comorbidities*	FALSE	58 (33.3)	12 (24.5)	-	-	-
	TRUE	116 (66.7)	37 (75.5)	1.54 (0.77–3.29, *p* = 0.241)	1.76 (0.78–4.21, *p* = 0.186)	6.96 (0.66–73.31, *p* = 0.104)
*History of definite or probable myocardial infarction*	FALSE	156 (89.7)	47 (95.9)	-	-	-
	TRUE	18 (10.3)	2 (4.1)	0.37 (0.06–1.34, *p* = 0.191)	-	0.06 (0.00–1.21, *p* = 0.066)
*History of congestive heart failure*	FALSE	165 (94.8)	44 (89.8)	-	-	-
	TRUE	9 (5.2)	5 (10.2)	2.08 (0.61–6.35, *p* = 0.208)	-	7.03 (0.25–199.13, *p* = 0.248)
*History of COPD*	FALSE	162 (93.1)	46 (93.9)	-	-	-
	TRUE	12 (6.9)	3 (6.1)	0.88 (0.19–2.91, *p* = 0.849)	-	0.91 (0.06–13.91, *p* = 0.948)
*History of diabetes*	FALSE	154 (88.5)	42 (85.7)	-	-	-
	TRUE	20 (11.5)	7 (14.3)	1.28 (0.48–3.12, *p* = 0.597)	-	1.69 (0.13–21.57, *p* = 0.683)
*Number of regular medications taken*	Mean (SD)	3.8 (2.8)	4.1 (2.6)	1.04 (0.92–1.16, *p* = 0.524)	0.97 (0.83–1.13, *p* = 0.726)	0.83 (0.55–1.24, *p* = 0.353)
*Researcher’s estimated risk of significant toxicity (CARG class)*	Low	26 (16.6)	8 (17.4)	-	-	-
	Low–medium	82 (52.2)	19 (41.3)	0.75 (0.30–2.01, *p* = 0.553)	-	0.41 (0.04–4.66, *p* = 0.458)
	Medium–high	44 (28.0)	18 (39.1)	1.33 (0.52–3.63, *p* = 0.562)	-	1.71 (0.05–55.17, *p* = 0.760)
	High	5 (3.2)	1 (2.2)	0.65 (0.03–4.88, *p* = 0.712)	-	7.44 (0.01–9018.15, *p* = 0.564)
*Researcher’s estimated risk of significant toxicity in percentage*	Mean (SD)	34.2 (20.2)	35.9 (19.4)	1.00 (0.99–1.02, *p* = 0.624)	-	0.92 (0.85–0.99, *p* = 0.036)
*AHQ: Presence of impaired hearing*	FALSE	136 (77.3)	37 (75.5)	-	-	-
	TRUE	40 (22.7)	12 (24.5)	1.10 (0.51–2.27, *p* = 0.796)	-	3.19 (0.46–22.19, *p* = 0.235)
*AHQ: Fall(s) in last 6 months*	FALSE	147 (83.5)	44 (89.8)	-	-	-
	TRUE	29 (16.5)	5 (10.2)	0.58 (0.19–1.46, *p* = 0.283)	-	0.46 (0.05–4.51, *p* = 0.499)
*AHQ: Interference of social activities due to health*	All the time	6 (3.4)	4 (8.2)	-	-	-
	Most of the time	21 (11.9)	8 (16.3)	0.57 (0.13–2.73, *p* = 0.466)	-	0.51 (0.01–19.88, *p* = 0.718)
	Some of the time	25 (14.2)	9 (18.4)	0.54 (0.12–2.52, *p* = 0.413)	-	0.11 (0.00–7.74, *p* = 0.303)
	A little of the time	12 (6.8)	7 (14.3)	0.88 (0.18–4.44, *p* = 0.868)	-	0.32 (0.00–44.30, *p* = 0.636)
	None of the time	112 (63.6)	21 (42.9)	0.28 (0.07–1.18, *p* = 0.065)	-	0.13 (0.00–4.51, *p* = 0.257)
*AHQ: Ability to take own medications*	Without help	169 (96.0)	45 (91.8)	-	-	-
	With some help or reminders	7 (4.0)	3 (6.1)	1.61 (0.34–6.04, *p* = 0.503)	-	0.20 (0.00–13.74, *p* = 0.438)
	Unable	0 (0.0)	1 (2.0)	7,954,942.58 (0.00–NA, *p* = 0.986)	-	3257.64 (0.00–Inf, *p* = 0.999)
*AHQ: Effect of health on walking one block*	Limited a lot	8 (4.5)	4 (8.2)	-	-	-
	Limited a little	15 (8.5)	11 (22.4)	1.47 (0.36–6.66, *p* = 0.600)	-	5.19 (0.08–338.32, *p* = 0.436)
	No limitations	153 (86.9)	34 (69.4)	0.44 (0.13–1.74, *p* = 0.206)	-	2.67 (0.05–152.63, *p* = 0.632)
*AHQ: Weight loss in the past 3 months*	Yes	89 (52.4)	31 (67.4)	-	-	-
	No	81 (47.6)	15 (32.6)	0.53 (0.26–1.04, *p* = 0.071)	-	0.50 (0.07–3.62, *p* = 0.490)
*AHQ: Describe mobility*	Bed- or chair-bound	1 (0.6)	0 (0.0)	-	-	-
	Can get out, but does not	4 (2.3)	5 (10.2)	2,647,724.96 (0.00–NA, *p* = 0.987)	-	415,273,785,083.42 (0.00–Inf, *p* = 0.998)
	Gets out	171 (97.2)	44 (89.8)	545,028.76 (0.00–NA, *p* = 0.988)	-	1,644,914,644.46 (0.00–Inf, *p* = 0.998)
*AHQ: Decline in food intake in the past 3 months*	Severe decrease	29 (16.5)	8 (16.3)	-	-	-
	Moderate decrease	51 (29.0)	22 (44.9)	1.56 (0.64–4.15, *p* = 0.345)	-	6.76 (0.32–142.17, *p* = 0.207)
	No decrease	96 (54.5)	19 (38.8)	0.72 (0.29–1.89, *p* = 0.481)	-	2.22 (0.17–29.80, *p* = 0.540)
*Rockwood CFS*	Mean (SD)	2.4 (1.2)	3.0 (1.4)	1.38 (1.07–1.80, *p* = 0.012)	-	1.32 (0.42–4.14, *p* = 0.618)
*Patient Questionnaire: Marital Status*	Single	29 (16.7)	7 (14.3)	-	-	-
	Married	116 (66.7)	34 (69.4)	1.21 (0.51–3.23, *p* = 0.676)	-	2.94 (0.07–119.76, *p* = 0.556)
	Widowed	18 (10.3)	8 (16.3)	1.84 (0.57–6.11, *p* = 0.308)	-	5.46 (0.44–67.40, *p* = 0.183)
	Other	11 (6.3)	0 (0.0)	0.00 (0.00–∞, *p* = 0.989)	-	0.00 (0.00–Inf, *p* = 0.994)
*Patient Questionnaire: Living situation*	Alone	46 (26.4)	13 (26.5)	-	-	-
	With partner/family	124 (71.3)	34 (69.4)	0.97 (0.48–2.05, *p* = 0.935)	-	0.24 (0.01–10.12, *p* = 0.450)
	Carer for partner/family	4 (2.3)	2 (4.1)	1.77 (0.23–10.18, *p* = 0.536)	-	1.53 (0.00–479.14, *p* = 0.882)
	Other	0 (0.0)	0 (0.0)	-	-	-
*Patient Questionnaire: Is there someone to help you with shopping or appointments?*	Always	119 (68.8)	34 (69.4)	-	-	-
	Sometimes	17 (9.8)	7 (14.3)	1.44 (0.52–3.64, *p* = 0.455)	-	1.64 (0.18–15.14, *p* = 0.659)
	Never	17 (9.8)	2 (4.1)	0.41 (0.06–1.53, *p* = 0.251)	-	0.16 (0.01–4.44, *p* = 0.268)
	I do not need help	20 (11.6)	6 (12.2)	1.05 (0.36–2.69, *p* = 0.923)	-	1.11 (0.09–13.02, *p* = 0.934)
*Patient Questionnaire: Smoking status*	Current	15 (8.6)	3 (6.1)	-	-	-
	Ex-smoker	84 (48.3)	21 (42.9)	1.25 (0.37–5.75, *p* = 0.742)	-	7.67 (0.25–236.82, *p* = 0.233)
	Never	75 (43.1)	25 (51.0)	1.67 (0.50–7.62, *p* = 0.448)	-	24.74 (0.55–1108.66, *p* = 0.094)
*Patient Questionnaire: comparison of health to others of similar age*	Not as good	17 (9.9)	8 (16.3)	-	-	-
	As good	74 (43.0)	13 (26.5)	0.37 (0.13–1.07, *p* = 0.060)	-	0.92 (0.09–9.44, *p* = 0.945)
	Better	77 (44.8)	24 (49.0)	0.66 (0.26–1.80, *p* = 0.399)	-	8.80 (0.48–161.63, *p* = 0.139)
	Does not know	4 (2.3)	4 (8.2)	2.12 (0.41–11.28, *p* = 0.362)	-	1.65 (0.01–523.73, *p* = 0.855)
*Patient Questionnaire: Consideration of likelihood of side effect from chemotherapy will lead to hospital stay*	Unlikely	84 (48.6)	19 (39.6)	-	-	-
	Not very likely	74 (42.8)	22 (45.8)	1.31 (0.66–2.64, *p* = 0.437)	-	6.32 (0.72–55.11, *p* = 0.091)
	Quite likely	13 (7.5)	7 (14.6)	2.38 (0.80–6.67, *p* = 0.104)	-	13.70 (0.47–402.77, *p* = 0.127)
	Very likely	2 (1.2)	0 (0.0)	0.00 (NA–∞, *p* = 0.989)	-	0.00 (0.00–Inf, *p* = 0.998)
*Patient Questionnaire: Consideration of likelihood of side effect from chemotherapy will lead to hospital stay in percentage*	Mean (SD)	24.2 (23.2)	28.0 (23.2)	1.01 (0.99–1.02, *p* = 0.323)	-	1.01 (0.98–1.05, *p* = 0.557)
*Patient Questionnaire: How does the patient rate their health today in percentage*	Mean (SD)	67.6 (25.9)	65.2 (26.2)	1.00 (0.98–1.01, *p* = 0.563)	-	0.98 (0.95–1.02, *p* = 0.272)

**Table 3 cancers-17-03303-t003:** Model performance metrics from the test dataset of (a) logistic regression model, (b) LASSO regression model and (c) random forest model. Abbreviations: CI—Confidence Interval; PPV—positive predictive value; NPV—negative predictive value; AUC—area under the curve.

	**(a) Logistic Regression**							
	**Selected Clinical Variables**				**Significant Variables**			
	**CCA**	**Imp**	**Up-Sample**	**Imp with Up-Sample**	**CCA**	**Imp**	**Up-Sample**	**Imp with Up-Sample**
Accuracy	0.6000	0.6600	0.5600	0.6200	0.5800	0.6600	0.6400	0.6600
95% CI	0.4518–0.7359	0.5123–0.7879	0.4125–0.7001	0.4717–0.7535	0.4321–0.7181	0.5123–0.7879	0.4919–0.7708	0.5123–0.7879
*p*-value	0.9962	0.9616	0.9995	0.9912	0.9985	0.9616	0.9809	0.9616
**Balanced accuracy**	**0.5658**	**0.5197**	**0.5395**	**0.6075**	**0.6382**	**0.5768**	**0.5921**	**0.5768**
Sensitivity (recall)	0.5000	0.2500	0.5000	0.5833	0.7500	0.4167	0.5000	0.4167
Specificity	0.6316	0.7895	0.5789	0.6316	0.5263	0.7368	0.6842	0.7368
PPV (precision)	0.3000	0.2727	0.2727	0.3333	0.3333	0.3333	0.3333	0.3333
**NPV**	**0.8000**	**0.7692**	**0.7857**	**0.8276**	**0.8696**	**0.8000**	**0.8125**	**0.8000**
**AUC**	**0.5789**	**0.6360**	**0.6075**	**0.6184**	**0.6590**	**0.6524**	**0.6590**	**0.6546**
AUC 95% CI	0.3730–0.7849	0.4557–0.8162	0.4279–0.7871	0.4393–0.7976	0.4708–8471	0.4682–0.8366	0.4691–0.8489	0.4666–0.8427
	**(b) LASSO**							
	**Selected Clinical Variables**				**Significant Variables**			
	**CCA**	**Imp**	**Up-Sample**	**Imp with Up-Sample**	**CCA**	**Imp**	**Up-Sample**	**Imp with Up-Sample**
Accuracy	0.5800	0.2200	0.5400	0.5200	0.7000	0.7400	0.6800	0.7200
95% CI	0.4321–0.7187	0.1153–0.3596	0.3932–0.6819	0.3742–0.6634	0.5539–0.8214	0.5966–0.8537	0.5330–0.8048	0.5751–0.8377
*p*-value	0.9985	1.0000	0.9999	0.9999	0.8753	0.6977	0.9282	0.7987
**Balanced accuracy**	**0.5526**	**0.4298**	**0.5548**	**0.5132**	**0.6031**	**0.5724**	**0.6469**	**0.6447**
Sensitivity (recall)	0.6053	0.0263	0.5263	0.5263	0.7895	0.8947	0.7105	0.7895
Specificity	0.5000	0.8333	0.5833	0.5000	0.4167	0.2500	0.5833	0.5000
PPV (precision)	0.7931	0.3333	0.8000	0.7692	0.8108	0.7907	0.8438	0.8333
**NPV**	**0.2857**	**0.2128**	**0.2800**	**0.2500**	**0.3846**	**0.4286**	**0.3889**	**0.4286**
**AUC**	**0.5526**	**0.4298**	**0.5548**	**0.5132**	**0.6031**	**0.5724**	**0.6469**	**0.6447**
AUC 95% CI	0.3852–0.7200	0.3167–0.5429	0.3884–0.7212	0.3449–0.6814	0.4433–0.7629	0.4352–0.7095	0.4840–0.8099	0.4831–0.8064
	**(c) Random Forest**							
	**Selected clinical Variables**				**Significant Variables**			
	**CCA**	**Imp**	**Up-Sample**	**Imp with Up-Sample**	**CCA**	**Imp**	**Up-Sample**	**Imp with Up-Sample**
Accuracy	0.7200	0.7000	0.6400	0.7000	0.7000	0.7600	0.7000	0.7400
95% CI	0.5751–0.8377	0.5539–0.8214	0.4919–0.7708	0.5539–0.8214	0.5539–0.8214	0.6183–0.8694	0.5539–0.8214	0.5966- 0.8537
*p*-value	0.79873	0.87529	0.9809	0.8753	0.8753	0.57668	0.8753	1
**Balanced accuracy**	**0.4737**	**0.4605**	**0.4781**	**0.5175**	**0.4890**	**0.500**	**0.6031**	**0.6294**
Sensitivity (recall)	0.0000	0.0000	0.1667	0.1667	0.08333	0.0000	0.4167	0.4167
Specificity	0.9474	0.9211	0.7895	0.8684	0.89474	1.0000	0.7895	0.8421
PPV (precision)	0.0000	0.0000	0.2000	0.2857	0.2000	-	0.3846	0.4545
NPV	0.7500	0.7447	0.7500	0.7674	0.75556	0.7600	0.8108	0.8205
**AUC**	**0.4737**	**0.5746**	**0.4781**	**0.5033**	**0.4890**	**0.5000**	**0.6031**	**0.6557**
AUC 95% CI	0.4377–0.5097	0.368–0.781	0.3499–0.6063	0.3789–0.6276	0.3936–0.5845	0.500–0.500	0.4433–0.7629	0.4819–0.8295

## Data Availability

As per the TOASTIE protocol [19], all information collected during the course of the TOASTIE study will be kept strictly confidential. Information will be held securely on paper and electronically. The research team will comply with all aspects of the 2018 Data Protection Act. This study is conducted within the framework of information governance (IG) good practice. The full code for all analyses can be found via the link in Appendix A.

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
