# Peer review of "CHEcking Diagnostic Differential Ability of Real Baseline Variables and Frailty Scores in Tolerance of Anti-Cancer Systemic Therapy in OldEr Patients (CHEDDAR-TOASTIE)"

_cancers, 2025, doi:10.3390/cancers17203303_

Round 1
Reviewer 1 Report
Comments and Suggestions for Authors
The article states pertinent results and is mostly well composed, but some points need to be refined to become more rigorous. More description in the discussion can help strengthen the clinical applicability of the findings, and subgroup stability and reporting consistency will contribute to improving the trustworthiness of the presentation. Alternative approach: Tackling the issue of methodological transparency in terms of up-sampling and pre-empting and equal formatting of the data across the manuscript will promote the quality and readability of the manuscript.
- In the Discussion section, discuss further in finer detail how models with high negative predictive values (e.g., 87% in the logistic regression model) can be applied in practice. Give 1-2 concrete instances of how this can be used to make shared decisions, like: not to reduce doses in low-risk patients or to be used with other geriatric evaluations such as those suggested by SIOG.
- In the Results (Section 3.2), there are the broad confidence intervals or infinity/NA values in the odds ratios (e.g. of high bilirubin or of some types of cancer) because they may be unstable due to small subgroups. It should be flagged in the table caption, and the subgroup conglomeration should be considered where possible.
- Add a statement discussing the possible bias brought about by up-sampling, such as overfitting to instances of the minority class that are duplicated. Make it clear that this process was only used on the training data and mention how validation helped to reduce the risk.
- The manuscript contains appropriate inconsistencies, including the capitalization of an abbreviation (e.g., CARG vs. Carg). This allows consistent reporting of SD in tables and checking that all supplementary file references (e.g., S1-S6) are correct and full.
Reviewer 2 Report
Comments and Suggestions for Authors
Dear authors;
Congratulations for your paper
It's a great effort working on improving elderly cancer patient's management
The only thing I have to mention is the fact that 65 years old is still the cutt off age for considering elder developed countries as yours
My only recommendation is to mention this (I'd rather prefer to consider elder for those with 70 or more years old)
However, congratulations again for your paper
Reviewer 3 Report
Comments and Suggestions for Authors
This study corresponds to research related to a well-known dataset (TOASTIE) for relevant analysis of chemotherapy against cancer in old subjects. However, some points should be clarified.
- The ten most relevant variables that were studied in this retrospective study should be clearly described.
- Figure 2 and table 2 are too long and difficult to understand, they should be simplified, particularly because this study gave no useful results.
- The conclusion is interesting for additional studies (P16-17) to obtain predictive useful data. Thus, this text could be shortened as this research did not produce valuable information.
Round 2
Reviewer 3 Report
Comments and Suggestions for Authors
The manuscript has been considerably improved. I recommend its publication.